# Freshwater Fish Biodiversity in a Large Mediterranean Basin (Guadalquivir River, S Spain): Patterns, Threats, Status and Conservation

Pedro Sáez-Gómez  and José Prenda *

Department of Integrative Sciences, University of Huelva, 21071 Huelva, Spain
*   Correspondence: jprenda@uhu.es

**Abstract:** The Guadalquivir River Basin is one of the largest in the Iberian Peninsula and has a remarkable freshwater biodiversity. Although many studies on hydrological regimes or water quality have been conducted in this basin the biodiversity of freshwater fish, as well as their distribution and conservation status, has never been globally addressed as in other Iberian basins. In this context, we synthesized information on freshwater fish using field procedures and a bibliographic search. Fish distribution patterns at different spatial scales and general environmental conditions were analyzed as well as the conservation status of the fish community. We documented the presence of 40 species (20 native and 20 exotic) in the basin during the 20th century until today. However, we only captured 18 species during the field sampling, with a prevalence for any native species of less than 23% (except *Luciobarbus sclateri*). The highest species richness was found in mid reaches, while the lower reaches had very low diversity values. Around 50% of species are threatened; according to the IUCN, several species are declining at an alarming rate and others are probably extinct and/or their current status is unknown. Human disturbances during the last few decades have caused serious changes in fish distribution and consequently to their conservation status. Hydrological alterations, intensive agriculture and introduced species are probably the principal reasons for Guadalquivir's ichthyofauna imperilment. Our study indicates an urgent and real need to identify important areas for fish conservation to guarantee a minimum fish biodiversity conservation over the long term, as well as effective strategies for fish recovery where it still is possible.

**Keywords:** fish fauna; Guadalquivir; distribution; ichthyofauna; Iberian Peninsula



## 1. Introduction

Freshwater ecosystems are the richest and most diverse ecosystems on earth and also one of the most threatened [1,2]. Their biodiversity is being reduced at an alarming rate, especially in arid and semi-arid regions where water demand for human uses is increasing (e.g., Mediterranean areas) [1,3–5]. At a European level, the conservation status of aquatic species is very poor, as 44% of freshwater mollusks, 37% of freshwater fishes and 23% of amphibians are threatened [6]. Freshwater fish are a basic element of this biodiversity, representing one-fourth of all living vertebrate species, the world's most endangered vertebrate group after amphibians and the most species-rich group among European vertebrates (546 fish species) [7–10]. European freshwater fish fauna consists of 12% critically endangered species, 10% endangered, 15% vulnerable and 4% species near-threatened. This is one of the highest levels of threat of any important taxonomic group ever evaluated in Europe [6,9,10].

In Mediterranean areas, where freshwater biodiversity is highly endemic [11], the World Conservation Union estimates that around 56% of endemic freshwater fish species are threatened: 18% critically endangered, 18% endangered and 20% vulnerable [5]. In this context, there is an urgent need for a permanently updated knowledge of the structure, di-

versity and distribution of Mediterranean freshwater fish communities due to the continual increase in human pressures on lands and resources in the Mediterranean (e.g., [12,13]).

The ecological status of the Guadalquivir River Basin is one of the least known in the Iberian Peninsula, despite its significant relevance in terms of area (around 10% of the Iberian land surface), biodiversity (one-third of Iberian fish species inhabit or have recently inhabited this basin; see results), impacts [14–17] and the high number of threatened species that inhabit it [11]. Before the 20th century, the fish fauna of the Guadalquivir was characterized, compared to other Iberian basins, by a diverse community of migratory species (catadromous and anadromous). These species constituted an important economic resource for many nearby towns [18]. However, the construction of reservoirs in the canal, as well as in the tributaries, was the main factor in the decline/extinction of these species [18]. Since the beginning of the 20th century, more than 150 reservoirs have been built in the basin, eleven of them in the main course of the river, which have isolated large sectors of the basin and blocked the migratory movements of many species [18]. One of the first inventories of species in the basin was carried out in 1989, which included a list of 19 native and introduced species with places and dates of capture in the basin [19]. However, the diversity, conservation status and distribution pattern of Guadalquivir's fish have rarely been addressed at a whole-basin level. Around 50% of native species recorded in the basin are threatened [10]. At a national scale, more than 85% of native species recorded in the Guadalquivir Basin are endangered [20]. At a regional scale, the information available is very scarce. The regional Red Data Book includes 15 endangered species and one that is extinct [21]. These data show the need for an urgent updating of the conservation status and threats to Guadalquivir's ichthyofauna; several species may be extinct nowadays and others are about to be so, while the introduction of exotic species continues to increase. A continuous monitoring program would also be necessary to assess population trends and to carry out real management and conservation strategies based on the actual status of this freshwater biodiversity component [22,23].

In this study, we examined the diversity, distribution and conservation status of the Guadalquivir River Basin's freshwater fish fauna. The specific objectives were: (1) to provide a complete list of the Guadalquivir fish fauna recorded from the 20th century to the present, (2) to determine current patterns of general fish distribution, richness and diversity, (3) to determine the conservation status of fish fauna in the basin and (4) to make recommendations for fish biodiversity conservation in the Guadalquivir River Basin.

## 2. Materials and Methods

### 2.1. Study Area

The Guadalquivir River is located in the south of the Iberian Peninsula, flowing west to the Atlantic Ocean. The main channel is 680 km in length, the fifth longest river in the Iberian Peninsula. The basin presents 80 main tributaries (basin areas between 12.6 km$^2$ and 8255.6 km$^2$) with a total drainage area of 57,439 km$^2$ and an average human population density of 69.6 hab/km$^2$ [24]. The basin has a typical Mediterranean hydrological regime with high intra- and inter-annual discharge variation [24–26].

The basin faces intense direct and indirect human pressures, mostly from agriculture [15,27], flow regulation (often related to agriculture), species introductions [28] and wastewater disposal ([29]; see Table 1). Agriculture has been undergoing major changes in Southern Spain since the end of the 1980s. A sharp change is taking place: from traditional non-irrigated extensive agriculture of typical Mediterranean crops (wheat, olives and wine) to a new, intensive, industrial and irrigated agricultural model, with high soil erosion levels, water abstraction and/or flow regulation, pesticide disposal and many other side effects (Table 1). A main consequence of these recent agricultural changes, usually associated with an increase in irrigated olive groves, is the introduction of an extraordinary suspended solid-loading into the drainage network that causes extreme levels in water turbidity [17]. The values of suspended sediment concentration (600 mg L$^{-1}$) registered by Ruiz et al. [30] in the main river channel are among the highest known in the world, two and three times

greater than that of the Danube (326 mg L$^{-1}$) and Amazon (200 mg L$^{-1}$), respectively [17]. These values are mainly located in the lower Guadalquivir reaches. This is seriously affecting the fish community and its nursery function, a well-known phenomenon in the Guadalquivir estuary [31].

**Table 1.** Guadalquivir River Basin environmental characteristics (data source: Confederación Hidrográfica del Guadalquivir: http://www.chguadalquivir.es/demarcacion-hidrografica-guadalquivir#PlanHidrologicodelGuadalquivir2009-2015 (accessed on 23 July 2021). * 'Dehesa' is a particular Mediterranean ecosystem where man has removed bushes to improve the farm [32].

| | Count | hm$^3$/year | km$^2$ | % |
|---|---|---|---|---|
| **Environmental information** | | | | |
| Human population (nº inhabitants) | 4,141,635 | | | |
| Municipality (nº) | 429 | | | |
| Annual average rainfall (mm) (1942–2005) | 561 | | | |
| Annual average contribution | | 7043 | | |
| Average net provision (m$^3$/ha/year) | 2906 | | | |
| Olive water demand (m$^3$/ha/year) | 1500 | | | |
| **Environmental quality** (nº water mass) | | | | |
| Good or moderate ecological status | 252 | | | 56.9 |
| Poor or bad ecological status | 191 | | | 43.1 |
| **Chemical status (nº water mass)** | | | | |
| Good chemical status | | | | 46.1 |
| Bad chemical status | | | | 53.9 |
| Nº pollution discharges | 1719 | | | |
| **Water use** | | | | |
| Supply | | 436.41 | | 10.9 |
| Industry | | 36.26 | | 0.9 |
| Agriculture | | 3504.06 | | 87.4 |
| Energy | | 31 | | 0.8 |
| **Land use** | | | | |
| Irrigated crops | | | 8460 | 14.7 |
| Rainfed cultivation | | | 24,000 | 41.8 |
| Forests | | | 7140 | 12.4 |
| Scrublands | | | 10,000 | 17.4 |
| Dehesas * | | | 4000 | 7.0 |
| Grasslands | | | 1400 | 2.4 |
| Bare or sparsely vegetated land | | | 829 | 1.4 |
| Water (lagoons, marshes, reservoirs, etc.) | | | 880 | 1.5 |
| Unproductive soil (urban áreas, roads, etc.) | | | 730 | 1.3 |

The flow of the Guadalquivir Basin is fully regulated. Thus, the water supply cannot be increased, but demand continues to rise more or less uncontrollably [33]. The number of reservoirs has reached a maximum. About 9193 hm$^3$ of river flow is nowadays retained in 29 large reservoirs (>100 hm$^3$) and more than 140 smaller ones (<100 hm$^3$) [34]. There are also numerous cut-offs, channeling and dredging works to promote river traffic, especially in the lower reaches of the river [18,35]. As a consequence, the natural flow pattern of the

main channel has been strongly modified. With respect to ecological status, according to the European Water Framework Directive (WFD; 2000/60/EC), 43% of all inland water bodies have poor or low level quality ([24], Table 1).

The Guadalquivir Basin displays a strong environmental asymmetry with respect to both left and right margins. Geology, physiography, climate and human pressure all vary. The right margin is located within the Iberian Massif, a low relief mountain range covered with oak forests and dehesas and characterized by a very low human population density, mostly devoted to marginal mountain agriculture and especially extensive farming and cattle raising. This area has a high conservation value and most of it is protected as several natural parks. The left margin corresponds to Cenozoic basins and the Betic Mountain chain. This is a highly developed flat land covered by intensive agriculture (irrigated and non-irrigated). Human population density is very high, and human pressures, in general, are correspondingly high.

## 2.2. Data Collection

We obtained fish data via field sampling and a bibliographic search. Sampling was conducted at 285 locations over the entire basin (216 in rivers and 69 in reservoirs) between June–September 2007 (reservoirs) and March–July 2008 (rivers) (Figure 1). The 285 sampling locations were distributed at random within 46 different sub-basins (including the main channel) that represented 56.3% of the total number of sub-basins and 96.1% of the total basin area, respectively (Figure 1).

Depending on location width and depth, two alternative sampling methods were used. A combination of different fishing methods is an appropriate option for sampling a wide range of habitats, species and fish sizes [36–39]. Typical stream sites with low salinity ($<1.5$ mS cm$^{-1}$) and shallow depth ($<1.2$ m) were electrofished by wading upstream along a channel length of 100 m during approximately one hour. The equipment comprised a control box delivering a pulsed direct current, 300/600 V, 4–6 A without block nets (Electracatch International, Honda EU 20i motor with a WF6 rectifier and a landing net with a 30 cm diameter and a 4 mm mesh size). The relative density was calculated for all captured species at all sampling points as catches per unit effort (hereafter, CPUE). The CPUE was defined as the number of individuals captured per 100 m stream length per hour sampling.

When the salinity and depth conditions did not allow electrofishing, mainly in large rivers and reservoirs, we used four types of passive traps: trammel nets, fyke nets, minnow traps and plastic bottle minnow traps. Two trammel nets (10 m × 2 m, 175 mm × 25 mm and 200 mm × 20 mm mesh size), three pairs of fyke nets (12 mm and 3 mm mesh size), fifteen metal minnow traps (6 mm mesh size) and ten pairs of plastic minnow traps (25 mm inlet) were set for a minimum of 8 h (see [38] for more details). After capture, fish were identified to species level, counted, measured and returned live to the river. The CPUE was defined here as individuals captured in the aforementioned trap combination (2 trammel nets + 3 pairs of fyke nets + 15 metallic mesh minnow traps + 10 pairs of plastic bottle minnow traps) per 12 h sampling.

A bibliographic search was performed for all fish species recorded in the Guadalquivir River Basin throughout the 20th century to develop a second presence/absence historical data set. This search covered scientific journals, technical reports, books and daily press with potential information on freshwater fish distribution. The Carta Piscícola Española (http://www.cartapiscicola.es accessed on 27 September 2022), and the International Standardization of Common Names for Iberian Endemic Freshwater Fishes [40] was used for scientific and common fish nomenclature, respectively.

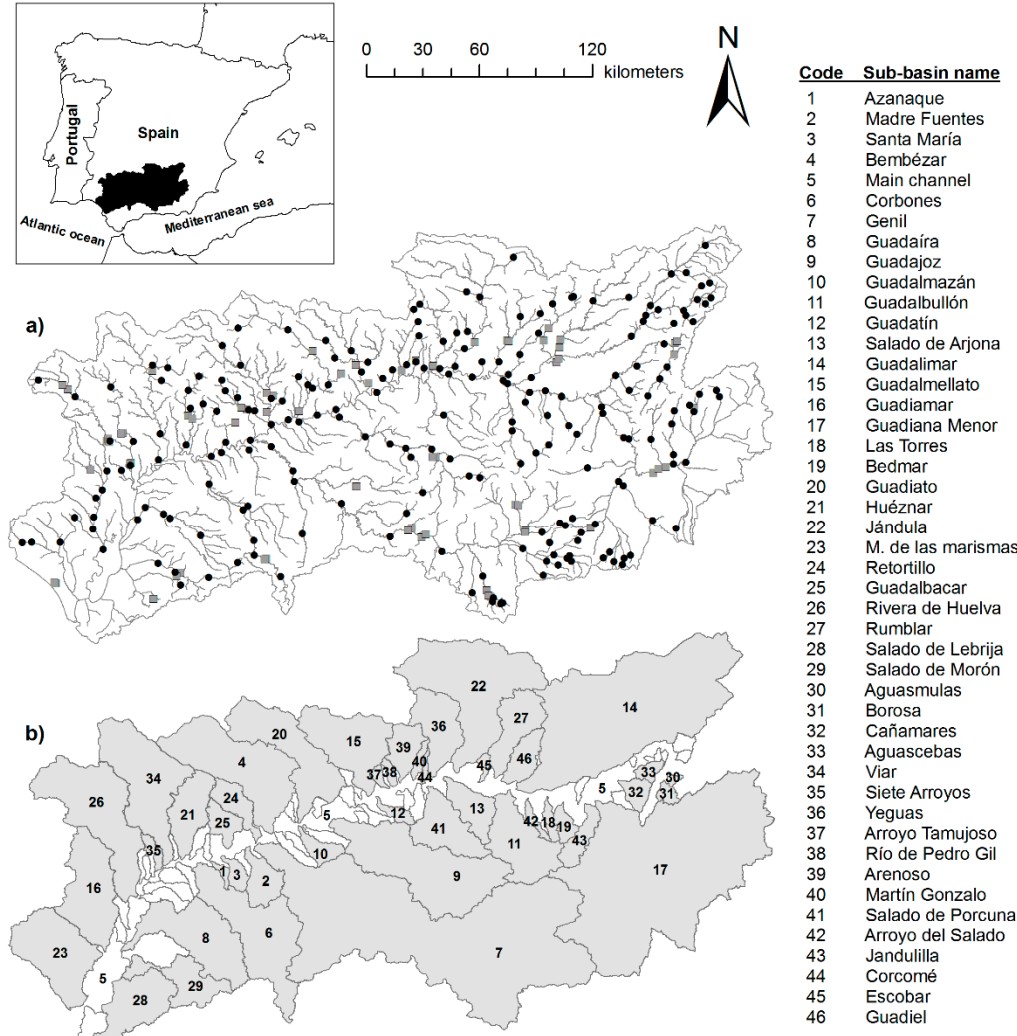

**Figure 1.** Map of the Guadalquivir River Basin. (**a**) Fish sampling sites: filled circles indicate rivers (n = 216) and grey squares denote reservoirs (n = 69). (**b**) Map of sub-basins sampled (gray color) and main channel (code-number 5) in the Guadalquivir River Basin. The code-number indicates the name for each sub-basin.

*2.3. Data Analysis*

Shannon (H') alpha diversity index (see [41]) and species richness values were compared at different scales and habitats using Kruskal–Wallis tests and Dunn's post test. All analyses were performed in SPSS Statistics® v. 21 [42]. The Shannon (H') index was calculated with PAST® (v.2.14) [43].

## 3. Results
### *3.1. Faunal Composition*

In an Iberian context, the Guadalquivir's ichthyofauna has great relevance as 31.1% of all native Iberian species live or have recently inhabited the Guadalquivir River Basin (Table 2), although it merely encompasses 9.8% of the total surface area of the Iberian Peninsula and only three families of native fish are not found in the basin. In particular, migratory species are very well represented, accounting for 54.5% of all Iberian species (Table 2). Cyprinidae is the richest family (both for native and exotic species) in the basin constituting 21.6% of Iberian species (Table 2) and 55.4% of the sampled species (Figure 2). Also, introduced species are widely represented in the Guadalquivir Basin with 48.4% of the total species recorded in Iberia (Table 2).

**Table 2.** Families and number (%) of Iberian freshwater fish species recorded in the Guadalquivir River Basin. Sources: Carta Piscícola Española for Spanish data (http://www.cartapiscicola.es, accessed on 27 September 2022) and Fishbase for Portuguese data (http://www.fishbase.org/Country/CountryChecklist.php?showAll=yes&c_code=620&vhabitat=fresh, accessed on 15 July 2022). * *Tinca tinca* is considered introduced according [44].

| | Family | Iberian Peninsula | Guadalquivir |
|---|---|---|---|
| Native | Petromyzontidae | 6 | 1 (16.7) |
| | Acipenseridae | 1 | 1 (100) |
| | Clupeidae | 2 | 2 (100) |
| | Anguillidae | 1 | 1 (100) |
| | Salmonidae | 2 | 1 (50) |
| | Cyprinidae | 37 | 8 (21.6) |
| | Cobitidae | 3 | 1 (33.3) |
| | Nemacheilidae | 1 | 0 (0) |
| | Cyprinodontidae | 2 | 1 (50) |
| | Valenciidae | 1 | 0 (0) |
| | Atherinidae | 1 | 1 (100) |
| | Gasterosteidae | 1 | 1 (100) |
| | Cottidae | 2 | 0 (0) |
| | Blenniidae | 1 | 1 (100) |
| | **TOTAL** | **61** | **19 (31.1)** |
| Introduced | Salmonidae | 4 | 1 (25.0) |
| | Esocidae | 1 | 1 (100) |
| | Cyprinidae | 11 * | 6 (54.5) |
| | Cobitidae | 1 | 0 (0) |
| | Nemacheilidae | 1 | 0 (0) |
| | Ictaluridae | 2 | 1 (50) |
| | Siluridae | 1 | 1 (100) |
| | Cyprinodontidae | 1 | 0 (0) |
| | Fundulidae | 1 | 1 (100) |
| | Poeciliidae | 3 | 1 (33.3) |
| | Cichlidae | 1 | 1 (100) |
| | Centrarchidae | 2 | 2 (100) |
| | Percidae | 2 | 0 (0) |
| | **TOTAL** | **31** | **15 (48.4)** |

During fieldwork a total of 18 fish species (9 native, 7 introduced and 2 translocated) were captured in the Guadalquivir River Basin (see Table S1 for sub-basin details). In addition, the bibliographic search revealed the presence of at least 22 additional species (11 native and 11 exotic) during the 20th century. Just over 18% of the sampled locations were apparently fishless (after 197.7 h spent electric fishing and 3578.7 m traversed at these points), all of them located in rivers (24% of all river locations). Cyprinidae was clearly the family with the greatest number of native and exotic sampled species, accounting in total for more than 50% of species richness (Figure 2). Species undetected during field sessions either have very low densities or they are extinct (see below).

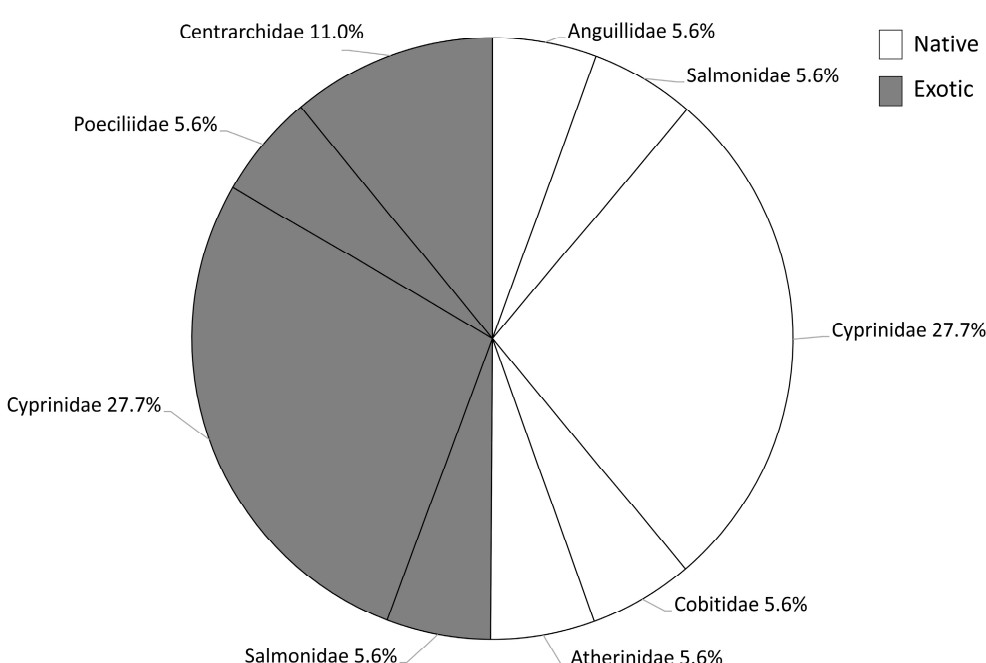

**Figure 2.** Percent of species number of the total families found in the Guadalquivir River Basin during the sampling sessions.

### 3.1.1. Sampled Species

Seven native primary freshwater fish species were captured (Table 3): brown trout (*Salmo trutta* Linnaeus, 1758), southern Iberian barbel (*Luciobarbus sclateri* (Günther, 1868)), Iberian arched-mouth nase (*Iberochondrostoma lemmingii* (Steindachner, 1866)), southern straight-mouth nase (*Pseudochondrostoma willkommii* (Steindachner, 1866)), calandino (*Squalius alburnoides* (Steindachner, 1866)), southern Iberian chub (*Squalius pyrenaicus* (Günther, 1868)), southern Iberian spined-loach (*Cobitis paludica* (de Buen, 1930)), and big-scale sand smelt (*Atherina boyeri* Risso, 1810), as well as one migratory species, European eel (*Anguilla anguilla* (Linnaeus, 1758)). The introduced exotic species captured included: rainbow trout (*Oncorhynchus mykiss* (Walbaum, 1792)), bleak (*Alburnus alburnus* (Linnaeus, 1758)), goldfish (*Carassius auratus* (Linnaeus, 1758)), common carp (*Cyprinus carpio* Linnaeus, 1758), eastern mosquitofish (*Gambusia holbrooki* Girard, 1859), pumpkinseed [*Lepomis gibbosus* (Linnaeus, 1758)] and largemouth bass (*Micropterus salmoides* (Lacépède, 1802)). The introduced translocated species (presumably from other Iberian river basins) were Pyrenean gudgeon (*Gobio lozanoi* Doadrio and Madeira, 2004) and a minnow species of unknown origin (*Phoxinus* spp.), probably the Pyrenean minnow (*Phoxinus bigerri* Kottelat, 2007) (Table 3). All native species had a very restricted distribution (prevalence less than 23%), except the southern Iberian barbel, which is the most widespread (a prevalence of 58.6%; see Table 3) and abundant species (more than 50% of the total captured fish) throughout the basin. Four species had prevalence values between 10% and 23%, and the four remaining species were present at less than 9% of the sampled sites. Excluding the southern Iberian barbel, the average prevalence of native fish was 9.8%. Among the locations with fishes, 42.1% presented at least one introduced species. *Atherina boyeri* is not a primary freshwater species; however, in the Guadalquivir basin there is an exclusively freshwater population (Zóñar pond) [20] and far from any contact with the coast (>100 km), so we decided to include it in this study.

**Table 3.** List of native and exotic freshwater fishes detected during the field sampling. Threat categories are referred to IUCN Red List [10] and Red book of Spanish freshwater fish, RBSF [20]. CR: Critically endangered; EN: endangered; VU: vulnerable; LC: least concern.

| | Family | Species | IUCN (2022) | RBSF (2001) | Prevalence (%) (n = 285) |
|---|---|---|---|---|---|
| **Native** | Anguillidae | *Anguilla anguilla* | CR | VU | 0.7 |
| | Salmonidae | *Salmo trutta* | LC | VU | 11.9 |
| | Cyprinidae | *Luciobarbus sclateri* | LC | LC | 58.6 |
| | | *Iberochondrostoma lemmingii* | VU | VU | 2.1 |
| | | *Pseudochondrostoma willkommii* | VU | VU | 22.8 |
| | | *Squalius alburnoides* | VU | VU | 15.8 |
| | | *Squalius pyrenaicus* | - | VU | 8.1 |
| | Cobitidae | *Cobitis paludica* | VU | VU | 15.4 |
| | Atherinidae | *Atherina boyeri* | LC | VU | 1.8 |
| | **TOTAL** | | | | **76.5** |
| **Introduced** | Salmonidae | *Oncorhynchus mykiss* | | | 1.8 |
| | Cyprinidae | *Alburnus alburnus* | | | 8.1 |
| | | *Carassius auratus* | | | 3.5 |
| | | *Cyprinus carpio* | | | 22.1 |
| | | *Gobio lozanoi* | | | 0.4 |
| | | *Phoxinus* spp. | | | 0.4 |
| | Poeciliidae | *Gambusia holbrooki* | | | 16.1 |
| | Centrarchidae | *Lepomis gibbosus* | | | 18.6 |
| | | *Micropterus salmoides* | | | 23.5 |
| | **TOTAL** | | | | **42.1** |

Largemouth bass and common carp were the most common exotic species (23.5% and 22.1% of the sampled sites, respectively,) followed by pumpkinseed (18.6%) and eastern mosquitofish (16.1%). The other species had prevalence values below 10% (Table 3). There are marked differences between native vs. introduced species prevalence in lotic environments, where the first are largely distributed (71.8% and 25.0%, respectively). Lentic environments, on the contrary, do not differ greatly, showing a prevalence of 91.3% for native species and 95.7% for introduced ones.

3.1.2. Bibliographic-Recorded Species

A total of 11 native species had been previously recorded in the Guadalquivir River Basin but were not captured in this study (Table 4), including 7 sedentary species: Oretanian arched-mouth nase (*Iberochondrostoma oretanum* (Doadrio and Carmona, 2003)), jarabugo (*Anaecypris hispanica* (Steindachner, 1866)), Iberian long-snout barbel (*Luciobarbus comizo* (Steindachner, 1864)), bogardilla (*Squalius palaciosi* Doadrio, 1980), three-spined stickleback (*Gasterosteus aculeatus* Linnaeus, 1758), freshwater blenny (*Salaria fluviatilis* Asso y del Rio, 1801), and baetican toothcarp (*Aphanius baeticus* (Doadrio, Carmona and Fernández-Delgado, 2002)); and 4 migratory species: Atlantic sturgeon (*Acipenser sturio* Linnaeus, 1758), sea lamprey (*Petromyzon marinus* Linnaeus, 1758), allis shad [*Alosa alosa* (Linnaeus, 1758)] and twaite shad [*Alosa fallax* (Lacépède, 1803)].

Some of these species may have gone extinct locally (e.g., Atlantic sturgeon, allis shad, three-spined stickleback and bogardilla) [18,20,45]. Others have been recently described (e.g., baetican toothcarp, Oretanian arched-mouth nase) [46,47] or have a very small distribution range (e.g., jarabugo) [48]. Iberian long-snout barbel and freshwater blenny records, according to Doadrio [20], have to be considered with caution and need to be checked, see [20].

Other records for 11 additional exotic species were obtained in the bibliographic search: tench (*Tinca tinca* (Linnaeus, 1758)), mummichog (*Fundulus heteroclitus* (Linnaeus, 1766)), black bullhead (*Ameiurus melas* (Rafinesque, 1820)), European catfish (*Silurus glanis* Lin-

naeus, 1758), Siberian sturgeon (*Acipenser baerii* Brandt, 1869), Adriatic sturgeon (*Acipenser naccarii* Bonaparte, 1836), northern pike (*Esox lucius* Linnaeus, 1758), oscar (*Astronotus ocellatus* (Agassiz, 1831)), chameleon cichlid (*Australoheros facetus* (Jenyns, 1842)), pirapitinga (*Piaractus brachypomus* (Cuvier, 1818)) and *Hypostomus* sp. (Table 4). Some of these species have been well established in the basin and surrounding areas for many years (e.g., mummichog, tench) [44,49], whereas others were recently introduced (e.g., black bullhead, European catfish) [50,51]. The probable historical presence of the Adriatic sturgeon has been a matter of debate, leading to several studies to either confirm or reject its occurrence [52–54], but there is still no consensus within the scientific community. However, some Adriatic sturgeon (>25 kg weight) that escaped from fish farms have been captured by anglers (authors' unpublished data). The status of this, and the rest of the introduced species that have been cited in the literature, is currently unknown.

**Table 4.** List of native and exotic freshwater fishes detected during the bibliographic search. Threat categories are referred to IUCN Red List [10] and Red book of Spanish freshwater fish, RBSF [20]. CR: critically endangered; EN: endangered; VU: vulnerable; LC: least concern. References for each data source ('Ref') are included in the reference list at the end of the article. (*) indicates the species that are still present in the Guadalquivir River Basin. (?) indicates species without data on their current status. *Iberochondrostoma oretanum* is not included in RBSF (2001).

| | Family | Species | IUCN (2022) | RBSF (2001) | Ref. | Current Presence | Locally Extinct |
|---|---|---|---|---|---|---|---|
| Native | Petromyzontidae | *Petromyzon marinus* | LC | EN | [18,20,55] | * | |
| | Acipenseridae | *Acipenser sturio* | CR | CR | [18,20,52, 54,56,57] | | * |
| | Clupleidae | *Alosa alosa* | LC | VU | [18,58] | | * |
| | | *Alosa fallax* | LC | VU | [18,20,59] | * | |
| | Cyprinidae | *Iberochondrostoma oretanum* | CR | - | [47] | * | |
| | | *Anaecypris hispanica* | EN | EN | [48] | * | |
| | | *Luciobarbus comizo* | VU | VU | [20] | | * |
| | | *Squalius palaciosi* | CR | EN | [20,60] | | ? |
| | Cyprinodontidae | *Aphanius baeticus* | EN | EN | [20,46,58] | * | |
| | Gasterosteidae | *Gasterosteus aculeatus* | LC | EN | [45,61] | | * |
| | Blenniidae | *Salaria fluviatilis* | LC | EN | [20] | | ? |
| Introduced | Acipenseridae | *Acipenser baerii* | | | [56] | ? | |
| | | *Acipenser naccarii* | | | [52,54] | * | |
| | Esocidae | *Esox lucius* | | | [62] | * | |
| | Cyprinidae | *Tinca tinca* | | | [20,44] | * | |
| | Ictaluridae | *Ameiurus melas* | | | [50,59] | * | |
| | Siluridae | *Silurus glanis* | | | [51] | * | |
| | Fundulidae | *Fundulus heteroclitus* | | | [20,59,63] | * | |
| | Cichlidae | *Astronotus ocellatus* | | | [64,65] | ? | |
| | | *Australoheros facetus* | | | [50] | * | |
| | Characidae | *Piaractus brachypomus* | | | [66,67] | ? | |
| | Loricariidae | *Hypostomus* sp. | | | [18] | ? | |

### 3.2. Patterns of Species Richness and Diversity

The total fish species richness (both native and exotic) for a given location ranged between 0 and 6 while the Shannon diversity index (H') ranged from 0–1.54 to 0–1.35 for native and exotic species, respectively, (see Table S2 for sub-basin details). A positive correlation was found between native and exotic richness and H' ($r_s = 0.178$, $p < 0.01$ and $r = 0.150$, $p < 0.05$, respectively). In addition, Wilcoxon signed-rank test showed no difference between native and exotic species in Shannon diversity ($Z = -1.36$, $N = 285$, $p > 0.05$), contrary to species richness ($Z = -3.90$, $N = 285$, $p < 0.001$). Therefore, sites with high values of native biodiversity tended to show similar values for exotic biodiversity due

to the high penetration of exotic species (overall prevalence of 42.1%; 95.7% for reservoirs and 25% for rivers).

Mean native biodiversity was independent of the type of water body, whether main channel, tributary or reservoir (Figure 3). However, mean exotic biodiversity was clearly overrepresented in reservoirs, which also acted as reservoirs for non-native fish, while they had a minor importance compared to native fauna in tributaries or the main channel (Figure 3). The right margin had higher biodiversity values than the left margin. Finally, with respect to Strahler's order, two different patterns emerged, one for native species and another for exotics. Native biodiversity peaked at order 2 and progressively was reduced until a minimum at locations higher than order 4, while exotics peaked at order 4 following a progressive increase from order 1 (Figure 3) (for all comparisons, Kruskal–Wallis, $p < 0.05$, Dunn's post test).

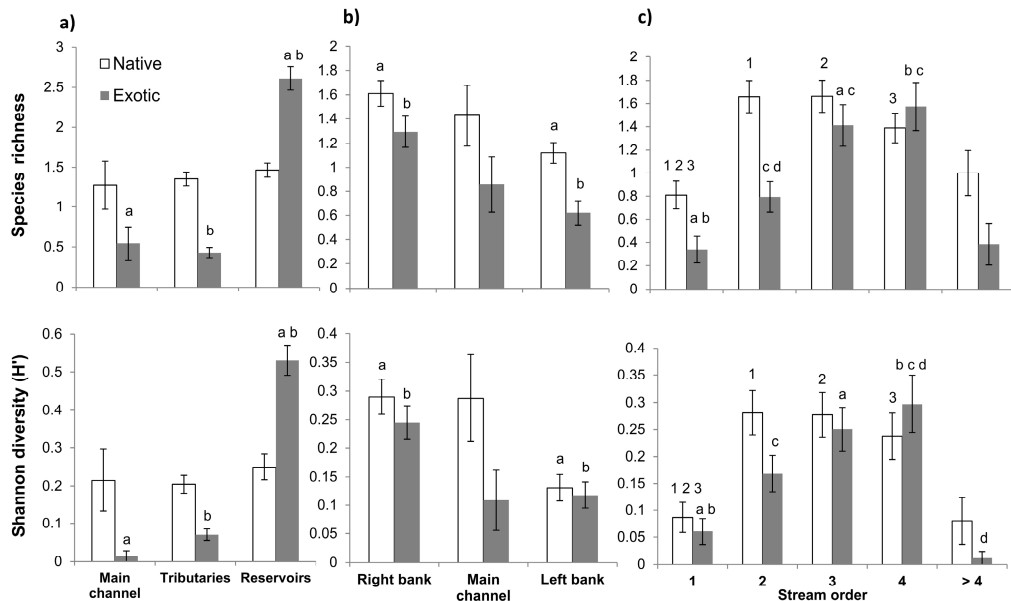

**Figure 3.** Main fish biodiversity descriptors (average species richness and Shannon H′ diversity index) for native and exotic species at different aquatic habitat typologies in the Guadalquivir Basin. Mainstream, tributaries and reservoirs (**a**); mainstream, left and right bank tributaries (**b**) and stream order (**c**), according to Strahler (1964). Error bars indicate standard error (SE). The same letter above bars shows that values are statistically different (Dunn's post test; $\alpha = 0.05$) when Kruskal–Wallis $p < 0.05$.

### 3.3. Conservation Status

To check the fish conservation status of the Guadalquivir River Basin, for each of 46 sub-basins the frequency of species included in any IUCN categories (Least Concern, Vulnerable, Critically Endangered) [10] was calculated (Figure 4). Among the total sub-basins sampled, 13% (n = 6) had no fish fauna and another one contained only exotic species (approximately 3% of total area). The rest of the sub-basins had a quite variable, but generally poor, conservation status (Figure 4). More than half of the sampled species were included in some of the most threatened IUCN categories (Tables 3 and 4). Similarly, 55% of species found in the bibliographical search were also included in some of these IUCN categories (Tables 3 and 4). The left margin presented a higher proportion for Least Concern and Vulnerable categories than the right one, but the right margin had more Critically Endangered species (Figure 4). Despite the existence of critically endangered species in the left bank (according to bibliographic search) they were not detected in the field study.

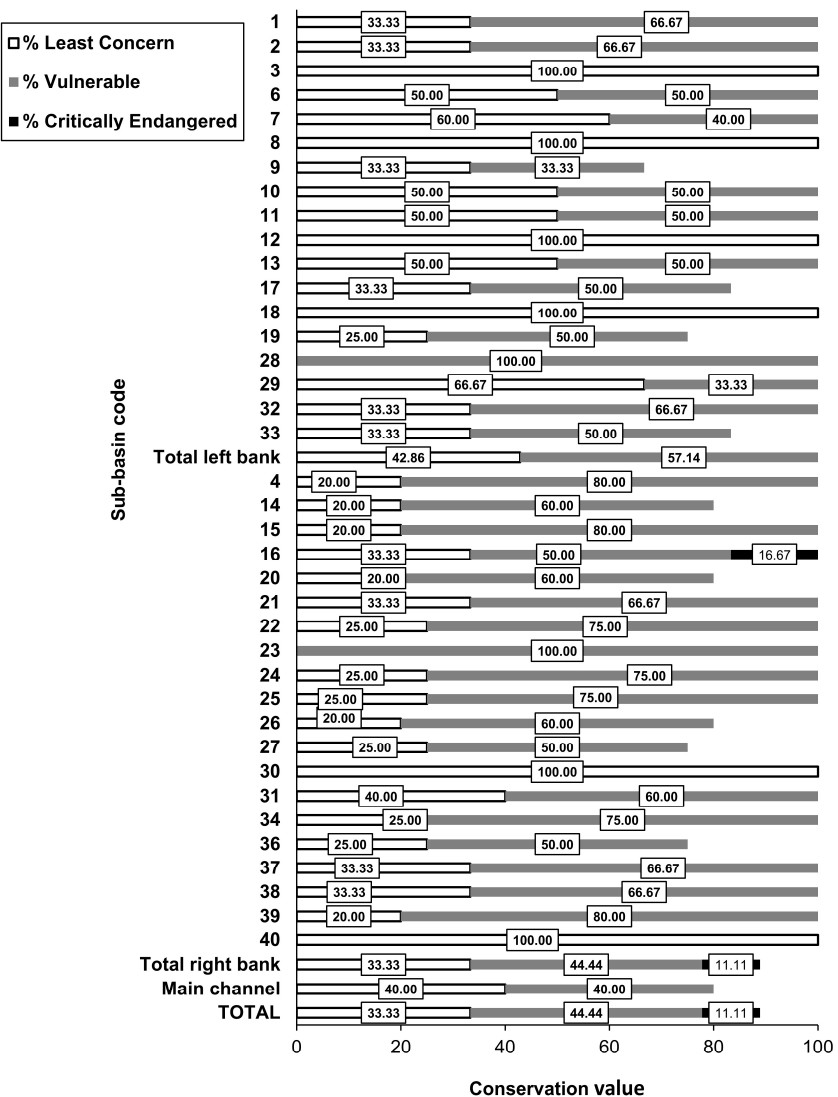

**Figure 4.** Conservation status (%) of sampled threatened fish species (IUCN red list criteria, version 15.1. July 2022) in different Guadalquivir River sub-basins.

## 4. Discussion

The Guadalquivir River Basin has a remarkable importance in the context of Mediterranean freshwater fish biodiversity and conservation [5]. Despite not being one of the most diverse basins in the region, it contains a high number of species, many of them endemic [5]. Around one-third of the native species registered in the Iberian Peninsula are nowadays present or have recently inhabited this basin and the species richness found within it is similar to other large Iberian basins (Figure 5).

Changes in the Guadalquivir Basin due to human activities during the last century have greatly modified the ecological characteristics of the water bodies and, subsequently, their fish communities [15,18,31]. During the 20th century, around 30 large dams were built and largely as a consequence of this the irrigated land increased by approximately 7000 km², 83% of the current total irrigated area [24]. This habitat transformation, especially reservoir building, has promoted the spread of exotic species [28] and has numerically reduced or driven to extinction a large proportion of native species, particularly migratory ones [18,70,71]. At the same time, agricultural intensification has markedly increased the discharge of agrochemicals (fertilizers and pesticides) as huge loads of suspended solids, which have greatly increased water turbidity, especially at the lower basin reaches [15,31]. Here, navigation has provoked strong changes in the main channel to reduce the distance

from Seville harbour to the sea by more than 70 km [72,73]. This has isolated the main channel from surrounding areas and has allowed the penetration of seawater further upstream, along with different marine species (e.g., *Dicentrarchus labrax*) [18,59].

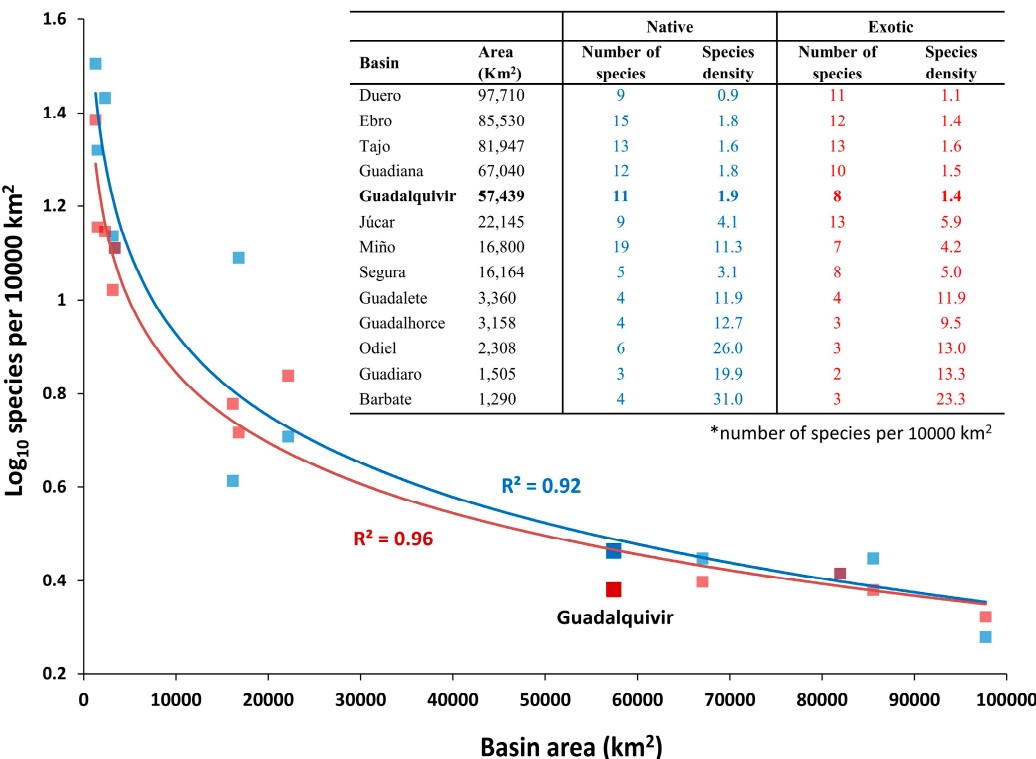

| Basin | Area (Km²) | Native Number of species | Native Species density | Exotic Number of species | Exotic Species density |
|---|---|---|---|---|---|
| Duero | 97,710 | 9 | 0.9 | 11 | 1.1 |
| Ebro | 85,530 | 15 | 1.8 | 12 | 1.4 |
| Tajo | 81,947 | 13 | 1.6 | 13 | 1.6 |
| Guadiana | 67,040 | 12 | 1.8 | 10 | 1.5 |
| **Guadalquivir** | **57,439** | **11** | **1.9** | **8** | **1.4** |
| Júcar | 22,145 | 9 | 4.1 | 13 | 5.9 |
| Miño | 16,800 | 19 | 11.3 | 7 | 4.2 |
| Segura | 16,164 | 5 | 3.1 | 8 | 5.0 |
| Guadalete | 3,360 | 4 | 11.9 | 4 | 11.9 |
| Guadalhorce | 3,158 | 4 | 12.7 | 3 | 9.5 |
| Odiel | 2,308 | 6 | 26.0 | 3 | 13.0 |
| Guadiaro | 1,505 | 3 | 19.9 | 2 | 13.3 |
| Barbate | 1,290 | 4 | 31.0 | 3 | 23.3 |

*number of species per 10000 km²

**Figure 5.** Species density–basin area relationship for the main Iberian basins. Drainage area, freshwater fish number and species density are shown. Raw data from: [28,68,69]. The blue color refers to the native species and the red to the exotic ones.

## 4.1. Guadalquivir Fish Biodiversity

Iberian freshwater fish are characterized by a small number of families but a high degree of endemism [20]. This may be a consequence of both geographic isolation and stressful ecological conditions derived from the extreme Mediterranean hydrologic regime [20,74]. This is characterized by autumn–winter catastrophic floods and summer droughts that leave many river sections dry during the summer and early autumn [25]. The native ichthyofauna of the Guadalquivir Basin is dominated, as in most Iberian basins, by the Cyprinidae family [69], which is well adapted to the extreme environmental conditions of the Mediterranean climate.

Stream fish community richness tends to increase from headwaters to lower sections of river basins [75–77]. However, this general biodiversity pattern is altered in the Guadalquivir Basin most likely as a response to poor habitat quality in the lower river sections, as has been observed in many highly populated basins in temperate areas [78,79]. Downstream reaches (usually >4 order) have high contamination levels, high prevalence of exotic species and physical habitat disturbances, among other things, that prevent a large number of species from establishing there [79,80].

The positive correlation observed between native and exotic fish richness may indicate that the dominant factors determining native diversity (e.g., competition, disturbance, availability of resources, etc.) are the same as those influencing invasions [81,82]. Here, sites with high values of native richness (a high habitat quality for a large number of species) are more vulnerable to invasion than those places with low diversity. This sharply contrasts with classical ecological theories that contend that diverse communities better resist exotic species invasion [83–85]. It may also be the case that the rivers draining the right margin



of the Guadalquivir had higher richness (and a fewer number of sites without fish) for both native and exotic fish than the rivers along the left margin. These latter rivers drain mostly flat, highly productive agricultural land occupied by intensive farming, industrial areas and densely populated human settlements, while those draining the right margin run through mountainous areas covered by natural forests, with low human disturbance and extensive agriculture and cattle raising [24,27].

*4.2. Introduced Species*

The introduction of exotic species in the Guadalquivir Basin is a general and widespread problem that is heavily threatening the native fauna. It is also causing the homogenization of freshwater biodiversity, a problem largely extending to all Iberian river basins [28,86,87].

Our results point out that most exotic biodiversity is a direct consequence of reservoirs. These human constructions provide the stable lentic habitats to which native fauna is not well adapted, but where, on the contrary, introduced species can thrive, thus acting as exotic species reservoirs, allowing them to disperse from here and to colonize other more or less nearby areas [28,79,88]. Without reservoirs, many of these exotic fish species would not survive in Mediterranean rivers or would do so in very low abundance [79,88]. The altitudinal segregation between native and exotic species is mainly caused by two factors. On the one hand, headwaters are usually isolated areas with low habitat transformation, in many cases included within protected areas [89]. On the other hand, many authors suggest that middle and lower river reaches have better conditions for the settlement of exotic species (slower water and a large cumulative number of reservoirs for downstream exotic fish dispersion) [28,79].

Many exotic species have been introduced and have subsequently proliferated in the Guadalquivir Basin during recent decades. Around 54% of these species have been introduced into Spain during the last century [90]. Fourteen of the detected introduced species (70% of the total) are now well established in the basin and/or are in the process of expansion. Four of these have colonized the basin within the last ten years: (1) bleak [91] and (2) black bullhead [50] are common in the middle and lower stretch of the basin, and finally (3) European catfish, the latest species detected in the basin [51], which involves a serious threat to regional wildlife [92]. The latter's preference for slow waters may favor its proliferation in the lowest sections of the basin, including the nature sanctuary of Doñana National Park and surrounding areas, with serious threats to this unique ecosystem.

*4.3. Conservation Status*

The conservation status of the Guadalquivir fish fauna is nowadays very poor, as more than half of the recorded native species in the basin are classified in a threat category, either nationally or internationally (see Tables 3 and 4).

The high degree of fish endemism in Mediterranean river basins, such as the Guadalquivir, demonstrates that this fauna, on average, displays rare and isolated populations naturally (e.g., oretanian arched-mouth nase, jarabugo, bogardilla and baetican toothcarp) [20,74]. Furthermore, some of these species have a low tolerance to disturbance, so that their extinction risk increases considerably [93,94]. Several formerly common species are now disappearing; for example, the Southern Iberian chub, the Iberian arched-mouth, the calandino and the European eel have considerably reduced distributions with respect to those observed in recent studies [20].

Two freshwater species are considered officially extinct in the Guadalquivir Basin: the three-spined stickleback and the Iberian long-snout barbel [18,59,95]. The first represented the southernmost population in Europe [61]. The direct causes of its extinction will never be known, but some speculate about habitat transformation, pollution and fish trading for aquarists [45]. The causes of Iberian long-snout barbel extinction are also unknown; however, the very small number of records and their antiquity [58] suggests that the historical presence of this species in the basin should be viewed with caution [95].

Migratory species are the worst preserved fish species. Only one of the five species (European eel, prevalence < 1%) with historical presence in the basin was captured, whereas two others are very scarce (i.e., sea lamprey and twaite shad) and the remaining two are regionally extinct (Atlantic sturgeon and allis shad [10,59]. The last published records for the sea lamprey come from 1992 and 1999, and in some studies it is considered extinct [59]; however, we are aware of sporadic, but periodic captures of this species in the estuary (authors' unpublished data). There are very few individual twaite shad dispersed in the lowest sections of the basin during early autumn and spring.

The relationship between the conservation of migratory fish fauna and the impact of dams is a widespread and well-known problem around the world [96,97]. Dams without adequate fish passage systems hinder access to upstream areas, which may include breeding grounds, this being one of the main challenges to their conservation [96,97]. Spain is one of the most dammed countries in the world, which represents an important environmental problem [28,70] because, among other reasons, most dams are impassable, i.e., it is not possible to install feasible fishways due to their height. In the Guadalquivir River, the Alcalá del Rio dam, built in 1931, is a major obstacle for upstream fish migration, especially for sturgeon [18]. Later on, in 1956, another dam (Cantillana dam) was built approximately 22 km upstream of Alcalá del Rio precipitating the total fragmentation of the lower section of the river from the middle one. Other specific causes involved in the regression of these species include: flow reduction (due to dams and irrigated crops), gravel extractions (damaging spawning grounds), pollution from urban and industry activities and overfishing, e.g., caviar industry, elver fishing [18,20,98].

*4.4. Conservation and Management Recommendations*

A regional strategy for freshwater fish conservation was approved in 2012 (see http://www.juntadeandalucia.es/boja/2012/60/boletin.60.pdf for more details, accessed on 26 September 2022) and its first actions, mainly species detection, range distribution estimates and ex situ programs, have recently been initiated. This strategy aims to act on species included in the regional Red Book: four critically endangered, five endangered and seven vulnerable [21]. However, three endangered species (*Salmo trutta*, *Alosa alosa* and *Alosa fallax*) are not included in the strategy. However, up to now, the current management and conservation measures taken have not seemed to alter the trend of the most endangered species due to the continuous degradation of the aquatic ecosystems of the basin [15,31,99] as well as the absence of large-scale actions that can interfere with human activities (e.g., agriculture) [15,31]. Only baetican toothcarp and brown trout, which have specific conservation plans for eight and six years, respectively, seem to have stabilized their populations [100,101].

The current conservation status (according to IUCN criteria) of all threatened fish species in the Guadalquivir Basin as well as introduced species should be revised and updated, placing each species in their appropriate global and regional coherent category, including possible extinctions. For example, the Andalusian regional Red Book includes the European eel in the 'Least Concern' category while in the IUCN Red list it appears as 'Critically Endangered'. As another example, the possible hybrid origin of bogardilla [20] should be clarified. This taxon is included in the regional conservation strategy and considerable funds could be invested for its conservation and protection, without even knowing its true taxonomic status. In addition, this "species" has not been collected in the last ten years [10].

The definition and detection of priority sites for freshwater fish conservation is necessary to determine the appropriate measures of restoration in the most valuable areas for fish. These should be areas with high native species richness or inhabited by endangered species. The legal declaration of protected natural areas would be useful to preserve them. Removing obsolete dams will increase connectivity among populations, thus avoiding their isolation [97] and promoting their recovery, especially among migratory species. This measure is being carried out in several countries in Europe and America [97]. The ex situ

programs could, in the future, provide individuals for repopulations, introductions and/or reintroductions for genetic improvement. In this regard, the development of live gene banks could help avoid the biodiversity loss of depauperated populations [20,102,103]. Finally, invasive species should be eliminated, if possible, especially from fish priority sites.

**Supplementary Materials:** The following supporting information can be downloaded at: https://www.mdpi.com/article/10.3390/d14100831/s1, Table S1: Species richness (S), number of families (F), Shannon (H') diversity indices and species density (D = n$^{\circ}$ species per 1000 km$^2$) for each sampled sub-basin (45 and the main channel). Table S2: List of freshwater fish species sampled in each sub-basin. Species name are codified using the first letter of the genus and the two first letters of the species.

**Author Contributions:** Conceptualization, P.S.-G. and J.P.; methodology, P.S.-G.; software, P.S.-G.; validation, P.S.-G. and J.P.; formal analysis, P.S.-G.; investigation, P.S.-G. and J.P.; resources, P.S.-G. and J.P.; data curation, P.S.-G.; writing—original draft preparation, P.S.-G.; writing—review and editing, P.S.-G. and J.P.; visualization, P.S.-G. and J.P.; supervision, J.P.; project administration, J.P.; funding acquisition, J.P. All authors have read and agreed to the published version of the manuscript.

**Funding:** This research was funded by the Junta de Andalucía, Convocatoria de Proyectos de Excelencia (P07-RNM-03309), and was carried out at the Centro Internacional de Estudios y Convenciones Ecológicas y Medioambientales (CIECEM) of the University of Huelva.

**Institutional Review Board Statement:** The authors declare that all procedures have been approved by the Andalusian Authority for Wildlife Protection. This study was carried out in accordance with national and international guidelines for care and use of animals.

**Data Availability Statement:** Data can be found within the paper and Supplementary Materials.

**Acknowledgments:** We wish to thank everyone from the CIECEM for their invaluable help and logistic support.

**Conflicts of Interest:** The authors declare no competing interests of financial or non-financial nature.

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
