# Peer review of "Freshwater Fish Biodiversity in a Large Mediterranean Basin (Guadalquivir River, S Spain): Patterns, Threats, Status and Conservation"

_diversity, doi:10.3390/d14100831_

Round 1

Reviewer 1 Report

The authors present a work, based on bibliography, but as well in a extensive field campaign, in order present the actual situation of diversity, distribution and conservation status of  Guadalquivir River Basin’s freshwater fish fauna, to analyze its evolution along time and implications for management.

The work is very interesting and it’s based on extensive literature complemented by a recent field survey. I have no criticism about the methodology used, which seems to me very solid. However, it could be interesting to relate the spread of exotic fish species and the decline of the native ones, particularly in relation to damming for intensive agriculture. River regulation is pointed out as the main cause of the low conservation status of the water bodies in this catchment, therefore it could be important to draw the evolution of fish assemblages with the construction of dams, of course this suggestion is only possible if there is available information along last decades.

I also think that the introduction needs to be better developed namely concerning the fish biodiversity and the existing threats of Guadalquivir catchment in the context of the Mediterranean area.

This work brings some important information, namely about the positive correlation observed between native and exotic fish richness may, where the authors only conclude that the dominant factors determining native diversity are the same as the ones influencing invasions. It’s true that the authors recognize that this pattern contrasts with classical ecological theories which generally conclude that in habitats with more diverse communities there is a higher resistance to exotic species invasions. However, these features should deserve also a deeper analysis.

The authors mention that the current management and conservation measures taken (the regional strategy approved in 2012) have not seemed to alter the trend of continuous degradation of this community. What are the causes of this failure?

Author Response

Reviewer's Comments:

REVIEWER #1:

General comments

The authors present a work, based on bibliography, but as well in a extensive field campaign, in order present the actual situation of diversity, distribution and conservation status of Guadalquivir River Basin’s freshwater fish fauna, to analyze its evolution along time and implications for management.

The work is very interesting and it’s based on extensive literature complemented by a recent field survey. I have no criticism about the methodology used, which seems to me very solid. However, it could be interesting to relate the spread of exotic fish species and the decline of the native ones, particularly in relation to damming for intensive agriculture. River regulation is pointed out as the main cause of the low conservation status of the water bodies in this catchment, therefore it could be important to draw the evolution of fish assemblages with the construction of dams, of course this suggestion is only possible if there is available information along last decades.

I also think that the introduction needs to be better developed namely concerning the fish biodiversity and the existing threats of Guadalquivir catchment in the context of the Mediterranean area.

This work brings some important information, namely about the positive correlation observed between native and exotic fish richness may, where the authors only conclude that the dominant factors determining native diversity are the same as the ones influencing invasions. It’s true that the authors recognize that this pattern contrasts with classical ecological theories which generally conclude that in habitats with more diverse communities there is a higher resistance to exotic species invasions. However, these features should deserve also a deeper analysis.

The authors mention that the current management and conservation measures taken (the regional strategy approved in 2012) have not seemed to alter the trend of continuous degradation of this community. What are the causes of this failure?

Response: we greatly appreciate your critical questions and suggestions to improve the quality of our manuscript. To assess changes over time in the expansion of exotic fish species and the decline of native ones, there are no robust continuous series of datasets. There are few field samples carried out in a representative area of the basin and in a relatively short time. The existing information corresponds to the XXI century and most of the basin's reservoirs were built between 1950 and 1970. However, we are currently working on historical bibliographic sources to try to obtain information on the fish distribution before the construction of dams and the establishment of exotic fish. Thank you very much for your suggestion regarding the analysis of the relationship between native and exotic richness. This is the subject of an independent study in which we are trying to quantify the environmental quality of different sections of the basin with different diversity of native and exotic species. Following their suggestions, we have expanded the information provided in the introduction about the biodiversity of fish in the Guadalquivir and its threats. Similarly, we have provided information on the inefficiency of conservation strategies.

Reviewer 2 Report

The manuscript is well written, the study presented very relevant. The authors just need to make minor changes following the corrections/comments in the uploaded document.

Author Response

REVIEWER #2:

General comments

The manuscript is well written, the study presented very relevant. The authors just need to make minor changes following the corrections/comments in the uploaded document.

Specific, minor comments:

L165 The sentence is not finished.

Response: ok, thank you. We have removed this half sentence that should have been deleted earlier.

L165-166: or is it 46 as indicated in the figure and mentioned again within the results?

Response: Nice point, thank you! You're right. There are 46 basins (including the main channel) as indicated in the list of sub-basins in Figure 1, where it can be seen that there are 45 sub-basins + main channel. We have changed '45' by '46 (including main channel)'.

L245: It should be mentioned in text before the figure.

Response: The ‘Figure 2’ is mentioned previously in the Line 221 “…of the sampled species (Figure 2). Also…”

L246: ‘either’ instead ‘must’.

Response: ok, we changed ‘must’ by ‘either’

L246: ‘they are’ instead ‘be’

Response: ok, we incorporate this suggestion. Now with this suggestion and the previous one, the sentence looks like this: “Species undetected during field sessions either have very low densities or they are extinct”.

Table 3: Names of families should not be in Italic.

Response:  Nice point, thank you! Now the family names are not in Italic.

Table 3: some parts of the table are not fully visible.

Response:  Thank you. The entire Table 3 appears now in the page 9.

L312: ‘that’ instead of ‘have

Response:  ok, changed.

L314: no comma here

Response:  ok, removed

L314: delete ‘it’

Response:  ok, removed

L318: not capital ‘C’

Response:  changed

L319: without ‘s’

Response:  ok, removed

Table 4: not fully visible

Response:  Now the entire Table 4 appears now in the page 11.

Table 4: families not in italic.

Response:  Right. Now the family names are not in Italic.

L339: patters?

Response:  We wanted to write ‘patterns’. We have changed 'patters' to 'patterns'

L355-356: As this division in 46 sub-basins is not visible in this Figure, I suggest to rephrase and connect first two sentences: "...Basin, for each of 46 sub-basens the frequency... ...and Critically Endangered (Figure 4)."

Response:  We appreciate the reviewer’s suggestion. Following the suggestion, we have rewritten the sentence:

“To check the fish conservation status the Guadalquivir River Basin, for each of 46 sub-basins the frequency of species included in any IUCN categories (Least Concern, Vulnerable, Critically Endangered) [10] was calculated (Figure 4).”

Reviewer 3 Report

The manuscript contains important information in the field. It is appropriate to publish after minor corrections. All minor corrections are noted in the paper as sticky notes.

Author Response

REVIEWER #3:

General comments

The manuscript contains important information in the field. It is appropriate to publish after minor corrections. All minor corrections are noted in the paper as sticky notes.

Specific, minor comments:

L31: keywords should not be the same as the title

Response: Right. We have changed the keywords following your suggestion.

L45: Should be given references

Response: We have now incorporated three references.

Table 2: All family names should not be italicized

Response: Right. Now the family names are not in Italic.

L253: Names of the author or authors describing the species should be given.

Response: Ok, all sampled species now incorporate the author.

Table 3: All family names should not be italicized

Response: Right. Now the family names are not in Italic.

L290: Names of the author or authors describing the species should be given.

Response: Ok, all bibliographic-recorded species now incorporate the author.

Table 4: All family names should not be italicized

Response: Right. Now the family names are not in Italic.

L365: change ‘S’ by ‘s’

Response: Done.

L515: Which are these species?

Response: These species are Salmo trutta, Alosa alosa and Alosa fallax, they are indicated in the section 3.1.1 and 3.1.2. However, we include here again its scientific name as it has been suggested.

L525: change ‘red by ‘Red’

Response: Done.
